# Skeletal Muscle MicroRNA Patterns in Response to a Single Bout of Exercise in Females: Biomarkers for Subsequent Training Adaptation?

**DOI:** 10.3390/biom13060884

**Published:** 2023-05-24

**Authors:** Alexandra Grieb, Angelika Schmitt, Annunziata Fragasso, Manuel Widmann, Felipe Mattioni Maturana, Christof Burgstahler, Gunnar Erz, Philipp Schellhorn, Andreas M. Nieß, Barbara Munz

**Affiliations:** 1Medical Clinic, Department of Sports Medicine, University Hospital Tübingen, Hoppe-Seyler-Str. 6, D-72076 Tübingen, Germany; 2Interfaculty Research Institute for Sports and Physical Activity, Eberhard Karls University of Tübingen, D-72074 Tübingen, Germany

**Keywords:** micro RNAs, skeletal muscle, exercise, biomarkers, females

## Abstract

microRNAs (miRs) have been proposed as a promising new class of biomarkers in the context of training adaptation. Using microarray analysis, we studied skeletal muscle miR patterns in sedentary young healthy females (*n* = 6) before and after a single submaximal bout of endurance exercise (‘reference training’). Subsequently, participants were subjected to a structured training program, consisting of six weeks of moderate-intensity continuous endurance training (MICT) and six weeks of high-intensity interval training (HIIT) in randomized order. In *vastus lateralis* muscle, we found significant downregulation of myomiRs, specifically miR-1, 133a-3p, and -5p, -133b, and -499a-5p. Similarly, exercise-associated miRs-23a-3p, -378a-5p, -128-3p, -21-5p, -107, -27a-3p, -126-3p, and -152-3p were significantly downregulated, whereas miR-23a-5p was upregulated. Furthermore, in an untargeted approach for differential expression in response to acute exercise, we identified *n* = 35 miRs that were downregulated and *n* = 20 miRs that were upregulated by factor 4.5 or more. Remarkably, KEGG pathway analysis indicated central involvement of this set of miRs in fatty acid metabolism. To reproduce these data in a larger cohort of all-female subjects (*n* = 29), qPCR analysis was carried out on *n* = 15 miRs selected from the microarray, which confirmed their differential expression. Furthermore, the acute response, i.e., the difference between miR concentrations before and after the reference training, was correlated with changes in maximum oxygen uptake (V̇O_2_max) in response to the training program. Here, we found that miRs-199a-3p and -19b-3p might be suitable acute-response candidates that correlate with individual degrees of training adaptation in females.

## 1. Introduction

The health benefits of physical activity and exercise are based on multiple morphological, physiological, and biochemical adaptation reactions of the whole body that are interconnected in a complex manner. Moreover, there is a wide range of individual reactions to a particular exercise regimen, and the factors that determine how a specific person reacts to a certain type of training still remain largely enigmatic. Biomarkers might be a useful tool to predict such individual reactions and to develop optimized, ‘individualized’, training regimens (for review, see [1]).

microRNAs (miRs) are short noncoding RNA molecules that regulate gene expression at multiple levels [2]. Some of them are more or less ubiquitously expressed, whereas others are specific for or highly enriched in particular tissues or cell types [3,4]. miRs-1, -133a/b, -206, -208a/b, -486, and -499a/b represent the group of so-called ‘myomiRs’ that are predominantly found in skeletal and/or cardiac muscle tissue [3,5,6,7,8]. They regulate myogenesis, skeletal muscle homeostasis, and skeletal muscle plasticity [9]. In particular, miR-1(-3p), miR-133a/b, miR-206, miR-486(-5p), and miR-499a(-5p) are involved in the regulation of myoblast proliferation and/or differentiation via a variety of mechanisms [10,11,12,13,14]. Similarly, miR-208b and miR-499a-5p promote conversion of muscle fibers towards a rather oxidative metabolism [7,15,16,17]. Moreover, miR-208b and miR-486(-5p) can promote muscle protein synthesis and hypertrophy [18,19,20], whereas miR-1(-3p), miR-133a, and miR-206 appear to be negative regulators of the latter [21,22,23,24].

Due to their widespread functions in the regulation of skeletal muscle plasticity, a central role in exercise adaptation has been attributed to myomiRs. Furthermore, various other miRs are known to regulate central aspects of muscle biology, such as cell proliferation and differentiation, protein synthesis and metabolism, which have been implicated in exercise adaptation.

However, despite the fact that several studies on circulating miRs and exercise adaptation in different settings have been published [25,26,27,28,29,30], little is known on skeletal muscle patterns in response to resistance and, particularly, endurance exercise. In addition, some of the data were only obtained in animal models; thus, their transferability to humans is questionable. Moreover, exercise protocols are highly heterogeneous, ranging from single bouts to training of several months’ duration. Finally, subjects’ characteristics are diverse, and, as in most areas of exercise biology, females are underrepresented in the relevant studies [31,32,33,34,35].

For skeletal muscle adaptation to endurance exercise, mitochondrial biogenesis and function are central. In response to acute endurance exercise, miR-31 and miR-23a have been shown to decline, leading to enhanced levels of PGC-1α in mice [36] and NRF1 in humans [37], both promoting mitochondrial biogenesis [38,39,40]. Furthermore, in mice, Yamamoto et al. [41] demonstrated reduced concentrations of miR-494 in murine skeletal muscle in response to a short-term endurance training program, leading to enhanced activities of TFAM (mitochondrial transcription factor A) and forkhead box protein J3 (FOXJ3), thus promoting mitochondrial biogenesis. Similarly, miR-378a-5p, miR-128(-3p), and miR-10a-5p were found to be induced after a single bout of aerobic exercise in healthy human subjects [42]. Both miR-378a species are transcribed from an area within the intron of the *PPARGC1b* gene, encoding PGC-1β [43]. On the other hand, miR-378a inhibits PGC-1β activity and fatty acid β oxidation, resulting in decreased energy availability [42,43,44,45]. Interestingly, induction of miR-378a-5p and miR-128(-3p) decreases MYOR (myogenic repressor) activity, thus promoting myoblast differentiation [46,47,48]. This mechanism might contribute to skeletal muscle repair and regeneration, e.g., after unaccustomed exercise, but also to training adaptation. Similarly, miR-9(5p), which decreases in response to exercise and is dysregulated in myopathies, and miR-152-3p, which is found at elevated levels in the circulation after exercise, might play a role in the regulation of myogenesis and muscle regeneration [37,49]. MiR-10a-5p, by contrast, besides repressing proinflammatory stimuli [50], enhances angiogenesis [51], thus promoting oxygen availability. This process might also be prone to regulation by other miRs, namely miR-15a(-5p), miR-16(-5p), and miR-126(-3p), all of which decrease VEGF levels [52,53,54]. Fernandes et al. [55] demonstrated reduced levels of miR-16(-5p) in response to long-term endurance training in rats, accompanied by increased VEGF concentrations and vascularization. Finally, miR-1(-3p), miR-133a/b, and miR-181a(-5p) have been described as being induced in skeletal muscle tissue in response to acute exercise [36,37,56]. This upregulation was paralleled by enhanced concentrations of MRF (myogenic regulatory factor) transcription factor family members, which are central to the regulation of skeletal muscle plasticity [36,37,56,57,58,59,60], suggesting the involvement of these miRs in training adaptation. However, interestingly, both miR-1(-3p) and miR-133a/b levels were found decreased in response to long-term endurance training [56,61], indicating that miR patterns and their regulation are dependent on training status.

Here, we analyzed skeletal muscle acute responses to a single bout of non-exhaustive cycling exercise in a group of young, healthy, sedentary females. Our aims were (I) to characterize skeletal muscle miR patterns in response to a single bout of exercise, using both a targeted and an untargeted approach, (II) to analyze the potential of miR microarray analysis of a small sample as an untargeted prescreening tool to identify candidate miRs for further testing in larger, ‘targeted’ qPCR studies, and (III) to test whether ‘acute-response’ miR patterns might be promising biomarkers in the context of developing individualized exercise regimens.

## 2. Materials and Methods

Subjects and study protocol. The analysis described in this paper was part of the ‘iReAct’ (individual response to physical activity) study that has previously been described [62]. Briefly, *n* = 42 (F: *n* = 30, M: *n* = 12) young (age 20–40 years), healthy, sedentary subjects underwent a structured training program on a bicycle ergometer. Due to the low number of males, only the group of female subjects was included in the current study, as already described [63]. Subjects’ baseline V̇O_2_max was 30.0 +/− 3.2 mL/kg·min; min: 24.2 mL/kg·min; max: 36.9 mL/kg·min. After enrollment, they were randomized to either start with a six-week period of moderate-intensity continuous training (MICT), followed by six weeks of high-intensity interval training (HIIT), or to accomplish the training program in reverse order. In both cases, subjects exercised three times per week. At baseline (T0), after the first six weeks of training (T1) and at the end of the program (T2), diagnostics, including spiroergometry to assess maximum oxygen uptake (V̇O_2_max), and a standardized one-hour reference training at submaximal intensity, calculated as the average of power outputs at lactate thresholds #1 and #2, was carried out. Samples employed for assessing skeletal muscle acute response in the context of this study were taken from this reference training (at T0). Detailed descriptions of the physiological results can be found elsewhere [64,65]. Changes in miR patterns at rest with training (delta T0-T1) and potential correlations between baseline miR patterns and training adaptation, namely V̇O_2_max, have previously been described [63].

Skeletal muscle biopsies. Before (pre) and two hours after the end of the reference training at baseline (post), fine-needle skeletal muscle biopsies were taken from the *m. vastus lateralis* as has previously been described [63].

miR isolation. Total RNA, including small RNA species, was isolated from skeletal muscle tissue using the miRNeasy Micro Kit (Qiagen, Hilden, Germany), according to the manufacturer’s instructions.

miR microarray analysis. Microarray analysis was carried out on samples obtained before and after the reference training from *n* = 6 female subjects. This group of subjects has previously been described [63] and was chosen based on maximum similarity with regard to sex, age, BMI, and baseline fitness. Affymetrix miR Array 4.0 analysis was carried out by ATLAS Biolabs, Berlin, Germany. Data were analyzed using Transcriptome Analysis Console (TAC), Version 4.0.2.15 (Thermo Fisher Scientific, Waltham, MA, USA). Based on the data format in this software, throughout this paper, individual miRs are referred to as ‘-3p’ or ‘-5p’ whenever possible.

Reverse transcription and qPCR analysis. Reverse transcription was carried out using the miRCURY LNA RT Kits (Qiagen, Hilden, Germany), according to the manufacturer’s instructions. For qPCR analysis, the miRCURY LNA miRNA PCR Assays (Qiagen, Hilden, Germany) in conjunction with primers specific for individual miRs, were employed. Due to technical issues, qPCR analysis was not possible with samples of one of the subjects; so, only data from *n* = 29 participants (instead of *n* = 30) could be included in the analysis. 

Statistical analysis. Statistical analysis was carried out using IBM SPSS Statistics Version 27.0 (IBM, Armonk, NY, USA). Data were tested for normal distribution using the Shapiro–Wilk test. For normally distributed data, mean values were compared between groups using t-tests for paired samples. Otherwise, Wilcoxon signed-rank tests were employed. Correlation analyses were carried out by means of the Spearman method, observing Cohen’s guidelines for evaluating the correlation coefficient r [66]. Based on the pilot, hypothesis-generating character of our study, Bonferroni correction of significance levels was not implemented. To test for concordance between microarray and qPCR data, we applied the ‘degree of concordance’ γ as defined by Goodman and Kruskal. All statistical tests were two-sided, with significance defined as *p* ≤ 0.05 (*; significant), 0.01 (**; very significant), or 0.001 (***; extremely significant).

KEGG (Kyoto Encyclopedia of Genes and Genomes) pathway analysis. KEGG pathway analysis was carried out using the TarBase v7.0 database within the online platform DIANA-miRPath v3.0 [67]. 

## 3. Results

Expression patterns of myomiRs and exercise-associated miRs was assessed by miR microarray analysis in a small cohort of subjects (*n* = 6) employing a targeted approach. To determine miR patterns in response to a single bout of endurance exercise, skeletal muscle biopsies were taken before and two hours after a 60-min non-exhaustive cycling session on a bike ergometer (‘reference training’) at baseline. First, in an initial screening approach, a small sample (*n* = 6) of all-female subjects that has previously been described [63] was screened by miR microarray analysis. Initially, we used a targeted approach to determine expression patterns of myomiRs and miRs that had previously been associated with skeletal muscle training adaptation. As shown in Figure 1A, we found that some myomiRs, specifically miR-1 (fold change −4.91; *p* = 0.008 **; the TAC software does not discriminate between miR-1-3p and -5p), 133a-3p (fold change −2.34; *p* = 0.041 *) and -5p (fold change −4.54; *p* = 0.006 **), -133b (fold change −2.76; *p* = 0.011 *), and -499a-5p (fold change −7.61; *p* = 0.046 *) were strongly and significantly downregulated in response to acute exercise. By contrast, other miRs, namely miR-206, -208a-3p and -5p, -208b-3p and -5p, -486-3p and -5p, -499a-3p, and -499b-3p and -5p remained unchanged (Figure 1A). When we analyzed miRs that had previously been described in the context of the skeletal muscle acute response to endurance exercise, we found that miRs-23a-3p (fold change −2.14; *p* = 0.004 **) and -378a-5p (fold change −1.99; *p* = 0.018 *) were significantly downregulated, whereas miR-23a-5p (fold change 2.2; *p* = 0.017 *) was significantly upregulated (Figure 1B). When we analyzed further miR species with a putative role in skeletal muscle exercise adaptation, we found significant downregulation of miR-128-3p (fold change −6.36; *p* = 0.018 *), but no significant regulation of miRs-107, -21-3p and -5p, 181a-3p, -9-3p, -9-5p, -31-3p, and -31-5p (Figure 1C). Finally, we analyzed the aerobic metabolism-associated miR species -10a-5p, -27a-3p, -126-3p, and -152-3p. Here, we found significant downregulation of miR-27a-3p (fold change −4.64; *p* = 0.007 **), miR-126-3p (fold change −3.94; *p* = 0.003 **), and miR-152-3p (fold change −7.82; *p* = 0.005 **) (Figure 1D).

Microarray expression patterns of miRs with fold changes ≥4.5 or ≤−4.5—untargeted approach. Next, we applied an untargeted strategy, screening our miR data for species with fold changes of ≥4.5 or ≤−4.5. Using this approach, we identified *n* = 55 miRs (*n* = 20 upregulated and *n* = 35 downregulated). Most fold changes were statistically significant (Appendix A).

Correlation analysis of microarray data with training adaptation. Next, we carried out Spearman correlation analysis for miR acute responses to exercise and aerobic training adaptation, namely gains in V̇O_2_max during the first six weeks of training (ΔV̇O_2_max T1). Thereby, as shown in Table 1, we identified *n* = 13 candidate miRs that correlated positively (r ≥ 0.6) and *n* = 6 that correlated negatively (r ≤ −0.6) with training adaptation. However, only four of these correlations (miR-19b-3p, r = 0.89, *p* = 0.019 *; miR-4667-5p, r = −0.94, *p* = 0.005 **; miR-6831-5p, r = −0.83, *p* = 0.042 *; and miR-7109-5p: r = −0.89, *p* = 0.019 *) reached statistical significance.

KEGG pathway analysis on miRs differentially expressed in microarray analysis. When we employed KEGG pathway analysis to identify targets of our set of *n* = 55 differentially expressed miRs, we specifically found that fatty acid biosynthesis and metabolism, ECM receptor interaction, and the Hippo signal transduction pathway were interesting potential targets (Table 2).

Concordance of microarray and qPCR analysis. To assess concordance between microarray and qPCR data, *n* = 19 miRs that were either significantly regulated during the acute response or whose expression patterns correlated with training adaptation were reassessed by qPCR analysis. The miRs for qPCR analysis were selected based on the following criteria: (I) strong and significant regulation in response to acute exercise (miR-1, miR-133a-3p and -5p, miR-133b, miR-499a-5p, and miRs-23a-5p, -378a-5p, and miR-27a-3p), (II) literature data suggesting a role in skeletal muscle plasticity, metabolism, and training adaptation (miR-1, miR-133a-3p and -5p, miR-133b, miR-499a-5p, miR-378a-5p, miR-27a-3p, miR-15a-5p, miR-18a-5p, miR-19b-3p, miR-132-3p, miR-155-5p, miR-199a-3p and -5p, and miR-497-5p), and (III) correlation of the acute response with subsequent training adaptation (ΔV̇O_2_max T1) (miR-15a-5p, miR-18a-5p, miR-19b-3p, miR-132-3p, miR-155-5p, miR-199a-3p and -5p, as well as miR-497-5p, miR-4330, miR-4743-5p, and miR-7151-3p, the latter three with unknown functions thus far). As shown in Table 3, we found that for means, the direction of regulation, i.e., ‘induction’ versus ‘repression’, was concordant between microarray and qPCR analysis in 73.33% of all cases. When we assessed concordance of data for individual subjects, we calculated the ‘Goodman and Kruskal’ degree of concordance γ. Thereby, we found that eight miRs showed a positive degree of concordance (γ > 0). MiR-199a-5p and -3p displayed a γ of 1, indicating perfect concordance of microarray and qPCR data (Figure 2). MiR-23a-5p, miR-4330, miR-4743-5p, and miR-7151-3p displayed a high degree of unspecific fluorescence in qPCR analysis and thus had to be excluded from the analysis.

qPCR analysis of miR expression patterns in a larger cohort of subjects (*n* = 29). To assess reproducibility of miR patterns in a larger cohort, samples of *n* = 29 female subjects, representing almost all the females included in the study (*n* = 30; one sample did not yield enough high-quality RNA) were analyzed by qPCR for the *n* = 15 miRs described in the preceding paragraph. As shown in Table 4 and Appendix A, we found that when this larger group of subjects was analyzed, general trends (upregulation versus downregulation) correlated between microarray and qPCR analyses for all 15 miR species analyzed, with most differences being significant or very significant. Furthermore, we found high degrees of correlation between expression patterns of individual miR species, specifically, three groups with similar regulatory patterns (group 1: miR-1-3p, miR-133a-3p, miR-133a-5p, miR-133b, miR-499a-5p, and miR-378a-5p; group 2: miR-497-5p, miR-199a-5p, and miR-27a-3p; and group 3: miR-15a-5p, miR-18a-5p, and miR-19b-3p) were identified (Table 5).

Correlation of qPCR data with ΔV̇O_2_max. Finally, we correlated training-induced adaptations, namely changes in V̇O_2_max, with miR patterns of all subjects (*n* = 29), based on qPCR-derived results. Subjects either started with MICT (*n* = 16) or HIIT (*n* = 13) and then switched to HIIT (*n* = 11) or MICT (*n* = 12), respectively. Subjects’ V̇O_2_max data have previously been published [64,65] and are summarized in Appendix A, indicating gains in relative V̇O_2_max throughout the training program. As shown in Figure 3, for ΔV̇O_2_max during the first training period, we found positive correlations with ΔmiR-199a-3p (r = 0.615; *p* = 0.025 *) in the HIIT group and with ΔmiR-19b-3p (r = 0.518; *p* = 0.04 *) in the MICT group. These two correlations had also been identified in the initial screening microarray analysis. Furthermore, additional correlations were found for acute changes in miR patterns and ΔV̇O_2_max during the second (T1/T2) or the entire (T2) training period; however, none of these reached statistical significance (data not shown).

## 4. Discussion

Our data demonstrate differential expression of multiple miRs in skeletal muscle tissue in response to a single submaximal bout of exercise.

In comparison to preexisting literature data, we demonstrated significant downregulation of myomiRs miR-1(-3p), miR-133a-3p and -5p, -133b, and miR-499a-5p, as well as of miR-19b-3p, -27a-3p, 199a-3p and -5p, 378a-5p, and 497-5p, both in the microarray (*n* = 6) and qPCR (*n* = 29) analyses. Some of these data were consistent with those of other studies that analyzed miR patterns in skeletal muscle in response to a single bout of aerobic exercise [36,37,42,56], whereas others were not. These differences are most likely due to different study species (animal–human, training status, health, and specifically sex, as most studies analyzing exercise responses were performed on male athletes), exercise types, intensities, and durations, as well as time points of species collection and methods of analysis, such as microarrays, RNAseq, or qPCR analysis, including normalization procedures. Timing, in particular, might be an important factor: Employing a ‘standard’ duration of 60 min per training session and obtaining just a single biopsy two hours after the reference training, we might have missed the differential expression of certain miRs as observed in other studies. Unfortunately, for obvious ethical and medical considerations, repeated muscle biopsies, i.e., ‘kinetics’ after exercise, are not an option. Functional analysis of differentially expressed miRs suggested their involvement in processes of angiogenesis (miR-10a-5p [51], miR-15a-5p [68], miR-18a-5p [69], miR-19b-3p [69,70], miR-126-3p [70], miR-132-3p [71], miR-155-5p [69], and miR-199a-3p/-5p [72]), mitochondrial biogenesis (miR-27a-3p [73]), inflammation (miR-155-5p [74,75], miR-10a-5p [50], and miR-29c-3p [76]), fiber type conversion (miR-27a-3p [77], miR-152-3p [78], and miR-30e-3p [79]), and the AKT-mTORC1 signal transduction pathway (miR-19b-3p [80,81], miR-128-3p [82], miR-126-3p [83], miR-1 [84], miR-133a [84], miR-199a-3p [85] and -5p [72], and miR-363-3p [86,87]). In addition, a broad variety of the miRs identified have been linked to processes of myogenesis, such as myoblast proliferation and differentiation (e.g., miR-19b-3p [88], miR-29c-3p [89,90], miR-30e-3p [79], miR-1 and miR-133a/b [91], miR-192-5p [92], and miR-660-5p [89], as well as miR-199a-5p, miR-128-3p, miR-155-p, and miR-499a-5p [93]). Furthermore, miR species miR-19b-3p might regulate skeletal muscle glucose metabolism by augmenting glucose uptake into the working muscle and promoting glucose oxidation [94]. Finally, in addition to its regulatory functions in the context of mitochondrial biogenesis [42,95], fatty acid oxidation [77], fiber type conversion [77], and muscle regeneration [96,97], miR-27a-3p appears to downregulate the activity of Runt-related transcription factor 1 (RUNX-1), which enhances erythropoiesis [61,98]. Thus, suppression of this miR might enhance oxygen supply and thus muscle aerobic capacity.

Moreover, KEGG pathway analysis using the DIANA TarBase v7.0 database suggested involvement of several induced (miR-155-5p, miR-1126-5p, and miR-2110) and repressed (miR-10a-5p, miR-15a-5p, miR-27a-3p, miR-29c-3p, miR-30e-5p, miR-199a-3p, miR-199b-3p, miR-363-3p, and miR-660-5p) miRs in fatty acid metabolism and biosynthesis. This might lead to enhanced aerobic capacity and more efficient regeneration [99,100]. Furthermore, KEGG pathway analysis indicated involvement of a group of repressed miRs (miR-15a-5p, miR-18a-5p, miR-19b-3p, miR-27a-3p, miR-29c-3p, miR-30a-3p, miR-30e-3p/-5p, miR-106-5p, miR-497-5p, miR-499a-5p, and miR-660-5p) in the regulation of both the FOXO and p53 signal transduction pathways. FOXO enhances switching of skeletal muscle metabolism from glucose to fatty acid oxidation by enhancing pyruvate dehydrogenase 4 (PDK4) activity. The tumor suppressor p53 maintains mitochondrial integrity and enhances mitochondrial biogenesis, thus being central in training-induced signal transduction [101,102]. In addition, p53-induced cell cycle arrest might allow repair of ROS-induced DNA damage in the context of muscle contraction [103,104]. Furthermore, p53 enhances fatty acid oxidation and oxidative phosphorylation [104,105,106], thus promoting adaptation to aerobic exercise. Finally, these data, indicating a broad-range regulation of miRs modulating fatty acid metabolism during the exercise acute response, are consistent with a previous publication [107], in which, employing a very similar exercise protocol, the authors found differential expression of a variety of genes encoding enzymes involved in fatty acid metabolism, of which, most interestingly, several are targets of the differentially expressed miRs identified in this study, for example the (mitochondrial) *HADH* (hydroxyacyl-coenzyme A dehydrogenase) gene, a central player in β oxidation, which, as depicted in the DIANA TarBase, is a target of (myo)miR-1-3p. Since Schild et al. [107] used a different methodology (proteomics instead of RNA-based approaches), this further validates our data, despite the fact that in the more distant future, more refined studies, employing different methods and assessing gene expression as well as miR and protein concentrations in the same set of samples, will have to be carried out. Moreover, KEGG pathway analysis also indicated involvement of the differentially expressed miRs (miR-10a-5p, miR-15a-5p, miR-18a-5p, miR-22-5p, miR-29c-3p, miR-30a-3p, miR-30e-3p, miR-192-5p, miR-199a-5p, and miR-497-5p) in the regulation of the Hippo signal transduction pathway [67]. Its effector protein YAP (yes-associated protein) enhances protein synthesis and muscle growth [108], but might also promote fatty acid oxidation and thus aerobic metabolic pathways [109]. However, since, on the other hand, MST-1 (mammalian sterile 20-like kinase), an inhibitor of YAP, has been shown to be inhibited by miR-199a-5p, functional consequences of these interrelationships will have to be more closely analyzed in the future. Finally, KEGG pathway analysis identified miRs involved in the regulation of ECM-receptor interactions (miR-19b-3p, miR-27a-3p, miR-199a-5p, and miR-497-5p), cell adhesion (miR-27a-3p, miR-199a-5p, and miR-497-5p), protein processing in the endoplasmic reticulum (miR-27a-3p and miR-497-5p), and cell cycle control (miR-18a-5p, miR-27a-3p, and miR-499a-5p). Since these processes play a role in the reestablishment of cellular integrity in response to training, these miRs might be of particular importance in the context of muscle regeneration after an acute bout of exercise (for review, see [100]).

Nevertheless, it is important to be aware of the fact that miRs form a complex network, with each miR targeting a multitude of genes and with each gene product being targeted by multiple miRs [110]. Thus, in the future, functional analyses, e.g., in knockout mice, are necessary to understand the complex interplay between skeletal muscle miRs and their targets in the context of training adaptation.

We could demonstrate strong inter-individual variability with regard to skeletal muscle miR acute responses, suggesting that miRs might be suitable biomarkers for individual training adaptation. Indeed, for HIIT, we could detect correlations between miR-199a-3p acute responses and training-associated ΔV̇O_2_max, and similar results were obtained for MICT with regard to miR-19b-3p. Nevertheless, the facts that (I) we did not apply corrections for multiple testing, due to the pilot character of our study, and (II) we were not able to establish correlations between miR acute patterns and training adaptations in the entire cohort, suggest that these correlations might not prove true in larger and/or more heterogeneous groups. Thus, in the future, more extensive studies and, specifically, direct comparisons between male and female athletes will have to be carried out. In addition, with regard to practicability, it would be desirable to switch to circulating miRs as markers, which, as described in several studies [25,26,29], will probably be even more complex, since these represent a mixture of miRs originating not only from skeletal muscle, but from a broad variety of tissues and organs.

In conclusion, our data suggest that microarray-based prescreening of a small subset of samples, followed by qPCR validation in the entire cohort, might be a valuable instrument for establishing skeletal muscle miR patterns and their modulation by exercise. Firstly, comparison of microarray and qPCR data yielded 73.33% concordance of individual trends (upregulation versus downregulation) in the six subjects initially analyzed. Secondly, when microarray data for miRs with fold changes of 4.5 or more were validated by qPCR in the entire cohort (*n* = 29), we even found that 100% of the general trends (means), i.e., average up- or downregulation as detected in the microarray analysis, could be confirmed for all miRs evaluated, with the restriction that, due to a high proportion of unspecific signals, qPCR data for miR-23a-5p, -4330, -4743-5p, and -7151-3p could not be analyzed.

In addition, in the future, miR-based screening of subjects prior to starting a training regimen might be a means to predict training responses, specifically in clinical settings but also in competitive sports. In particular, such a prediction might be helpful in pre-identifying subjects for which an insufficient response to ‘standard’ training protocols is likely. These individuals might then directly be allocated to alternative, yet-to-be-defined and ‘non-standard’ training regimens, from which they might potentially benefit to a higher degree. 

## Figures and Tables

**Figure 1 biomolecules-13-00884-f001:**
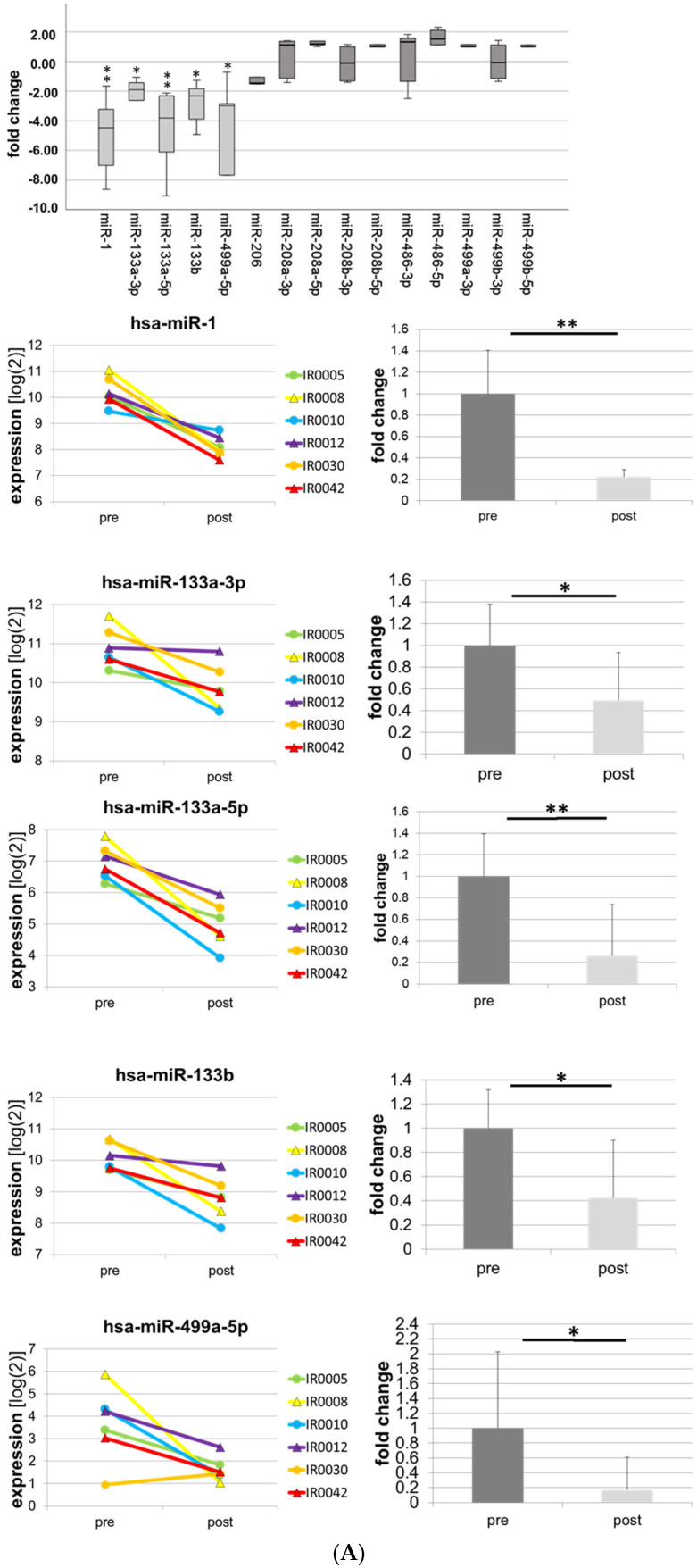
MiR acute responses as detected by microarray analysis. Box plots, spaghetti diagrams, and bar graphs show expression levels and fold changes in skeletal muscle acute responses in *n* = 6 subjects. (**A**) myomiRs miR-1, miR-133a-3p and -5p, and miR-133b and miR-499a-5p, (**B**) miR-23a-3p and -5p and miR-378a-3p and -5p, (**C**) miR-9-3p and -5p, miR-21-3p and -5p, miR-31-3p and -5p, miR-107, -128-3p, and -181a-3p and -5p, (**D**) miR-10a-5p, miR-27a-3p, miR-126-3p, and miR-152-3p. pre: before exercise, post: after exercise. * *p* ≤ 0.05, ** *p* ≤ 0.01.

**Figure 2 biomolecules-13-00884-f002:**
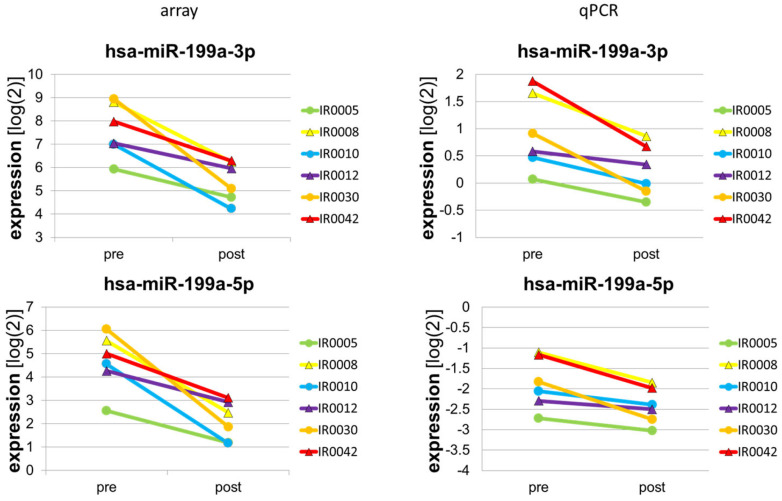
Regulation of miR-199a-3p and -5p in the skeletal muscle acute response, as assessed by microarray and qPCR analysis. Spaghetti plots show skeletal muscle expression levels of miR-199a-3p and -5p, pre and post exercise, as assessed by microarray and qPCR analysis in *n* = 6 subjects. Identical colors represent identical subjects.

**Figure 3 biomolecules-13-00884-f003:**
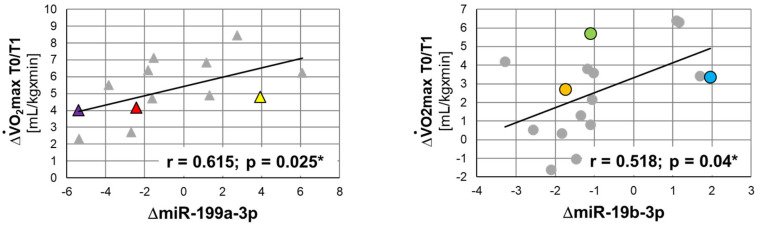
Correlations between skeletal muscle miR acute responses and training-associated changes in V̇O^2^max between T0 and T1. Regression analysis of acute responses of miR-199a-3p (left panel) and miR-19b-3p (right panel), as assessed by qPCR, and ΔV̇O_2_max between T0 and T1. Triangles represent subjects assigned to HIIT during the first training period, and circles represent those who started with MICT. Colored symbols represent subjects included in the initial microarray screening. Spearman correlation coefficients r and *p* values are indicated. * *p* ≤ 0.05.

**Table 1 biomolecules-13-00884-t001:** Spearman correlation coefficients between miR acute responses and ΔV̇O_2_max T1. Correlation coefficients > 0.7 or <−0.7 indicate strong correlation, and coefficients between 0.6 and 0.7 or −0.6 and −0.7 indicate moderate correlation. MiRs labeled in red were further tested in subsequent qPCR experiments. ΔV̇O_2_max T1: changes in maximum oxygen uptake [mL/kg·min] between T0 and T1. * *p* ≤ 0.05, ** *p* ≤ 0.01.

miR	Correlation Coefficient r (Spearman)	*p*
hsa-miR-15a-5p	0.77	0.072
hsa-miR-18a-5p	0.6	0.208
hsa-miR-19b-3p	0.89	0.019 *
hsa-miR-132-3p	0.6	0.208
hsa-miR-155-5p	0.66	0.156
hsa-miR-199a-3p	0.6	0.208
hsa-miR-199a-5p	0.6	0.208
hsa-miR-199b-3p	0.6	0.208
hsa-miR-497-5p	0.6	0.208
hsa-miR-3175	0.77	0.072
hsa-miR-4306	0.71	0.111
hsa-miR-7151-3p	0.77	0.072
hsa-miR-7641	0.71	0.111
hsa-miR-197-5p	−0.6	0.208
hsa-miR-4330	−0.6	0.208
hsa-miR-4667-5p	−0.94	0.005 **
hsa-miR-4743-5p	−0.77	0.072
hsa-miR-6831-5p	−0.83	0.042 *
hsa-miR-7109-5p	−0.89	0.019 *

**Table 2 biomolecules-13-00884-t002:** KEGG pathway analysis of miR acute responses in skeletal muscle tissue, based on microarray analysis. The prefix ‘hsa-’ indicates the respective KEGG pathway identification number within the *homo sapiens* database. The number of target genes in the respective pathways and the corresponding miRs are listed.

	Signaling Pathway	*p*	Number of Genes	miR	
1	prion diseases (hsa05020)	<1 × 10^−325^	18	-22-5p	-27a-3p
				-30a-3p	-30e-3p
				-106b-5p	-192-5p
2	fatty acid biosynthesis (hsa00061)	<1 × 10^−325^	5	-10a-5p	-15a-5p
				-27a-3p	-29c-3p
				-30e-5p	-199a-3p
				-199b-3p	-1226-5p
				-2110	
3	fatty acid metabolism (hsa01212)	<1 × 10^−325^	24	-10a-5p	-15a-5p
				-27a-3p	-29c-3p
				-30e-5p	-155-5p
				-199a-3p	-199b-3p
				-155-5p	-363-3p
				-660-5p	-1226-5p
4	ECM-receptor interaction (hsa04512)	<1 × 10^−325^	43	-22-5p	-19b-3p
				-27a-3p	-29c-3p
				-30a-3p	-30e-3p
				-101-5p	-199a-5p
				-497-5p	
5	proteoglycans in cancer (hsa05205)	<1 × 10^−325^	142	-15a-5p	-19b-3p
				-22-5p	-27a-3p
				-29c-3p	-30a-3p
				-30e-3p	-30e-5p
				-132-3p	-155-5p
				-106b-5p	-192-5p
				-199a-3p	-199a-5p
				-497-5p	
6	viral carcinogenesis (hsa05203)	9.99 × 10^−16^	123	-15a-5p	-18a-5p
				-19b-3p	-22-5p
				-27a-3p	-29c-3p
				-30a-3p	-30e-3p
				-30e-5p	-362-5p
				-2110	-2115-5p
7	Hippo signal transduction pathway (hsa04390)	1.11 × 10^−13^	100	-10a-5p	-15a-5p
				-18a-5p	-22-5p
				-27a-3p	-29c-3p
				-30a-3p	-30e-3p
				-106b-5p	-132-3p
				-192-5p	-199a-5p
				-497-5p	
8	lysine degradation (hsa00310)	2.06 × 10^−11^	33	-15a-5p	-18a-5p
				-22-5p	-27a-3p
				-29c-3p	-30a-3p
				-30e-3p	-30e-5p
				106b-5p	132-3p
				-192-5p	-197-5p
				-497-5p	-660-5p
				-1226-5p	-2110
				-2115-5p	
9	cell cycle (hsa04110)	1.65 × 10^−5^	84	-15a-5p	-18a-5p
				-27a-3p	-30a-3p
				-30e-3p	-30e-5p
				-101-5p	-106b-5p
				-132-3p	-192-5p
				-499-5p	
10	TGFβ signal transduction pathway (hsa04350)	3.88 × 10^−5^	51	-15a-5p	-18a-5p
				-19b-3p	-27a-3p
				-30a-3p	-30e-3p
				-106b-5p	-132-3p
				-155-5p	-199a-5p
				-497-5p	
11	steroid biosynthesis (hsa00100)	0.00016	12	-18a-5p	-29c-3p
				-30a-3p	-30e-3p
				-155-5p	-192-5p
				-199a-3p	-199b-3p
				-362-5p	-2110
12	adherens junction (hsa04520)	0.00037	52	-15a-5p	-22-5p
				-27a-3p	-30e-5p
				-106b-5p	-199a-5p
				-362-5p	-497-5p
13	p53 signal transduction pathway (hsa04115)	0.00038	46	-15a-5p	-18a-5p
				-19b-3p	-27a-3p
				-29c-3p	-30a-3p
				-30e-3p	-30e-5p
				-106b-5p	-497-5p
				-499a-5p	-660-5p
14	protein processing in the endoplasmic reticulum (hsa04141)	0.00059	109	-15a-5p	-22-5p
				-27a-3p	-30e-5p
				-106b-5p	-197-5p
				-497-5p	
15	Hepatitis B (hsa05161)	0.0029	88	-15a-5p	-19b-3p
				-27a-3p	-29c-3p
				-30a-3p	-30e-3p
				-106b-5p	-155-5p
				-497-5p	
16	chronic myeloic leukemia (hsa05220)	0.0131	53	-19b-3p	-27a-3p
				-29c-3p	-30a-3p
				-30e-3p	-101-5p
				-106b-5p	-155-5p
				-497-5p	

**Table 3 biomolecules-13-00884-t003:** Concordance between miR microarray and qPCR data. Summary of concordant and discordant data. Overall concordant data (mean upregulation or mean downregulation in both microarray and qPCR) are labeled in green, discordant data in red. The degree of concordance γ ((C-D)/(C + D); C: number of concordant pairs, D: number of discordant pairs) indicates concordance between the two methods for specific individuals: γ = 1: complete concordance, γ = 0: no statistical correlation, γ = −1: complete discordance).

miR	Microarray		qPCR		Concordance		
	Fold Change	*p*	Fold Change	*p*	C	D	γ
-1/-1-3p	−4.91	0.008	−1.54	0.203	5	1	0.66
-133a-3p	−2.34	0.041	−4.07	0.078	4	2	0.33
-133a-5p	−4.54	0.006	+1.21	0.032	1	5	−0.66
-133b	−2.76	0.011	−1.45	0.565	3	3	0
-499a-5p	−7.61	0.046	−1.33	0.249	4	2	0.33
-15a-5p	−5.12	0.021	−1.16	0.462	1	5	−0.66
-18a-5p	−7.39	0.028	1.0	0.576	1	5	−0.66
-19b-3p	−5.33	0.032	−1.04	0.45	2	4	−0.33
-27a-3p	−4.64	0.007	−1.78	0.016	5	1	0.66
-132-3p	+6.46	0.028	+1.27	0.614	4	2	0.33
-155-5p	+7.32	0.028	−1.16	0.731	2	4	−0.33
-199a-3p	−5.8	0.035	−3.78	0.127	6	0	1
-199a-5p	−7.69	0.026	−1.49	0.023	6	0	1
-378a-5p	−1.99	0.018	−1.29	0.01	5	1	0.66
-497-5p	−6.46	0.043	−1.29	0.206	3	3	0

**Table 4 biomolecules-13-00884-t004:** MiR acute responses in a larger cohort. In total, 15 miRs were selected from the microarray for qPCR analysis in a larger cohort of subjects (*n* = 29). Fold changes and *p* values are displayed as indicated. Overall, *n* = 13 miRs were downregulated, *n* = 1 (miR-155-5p) was upregulated, and *n* = 1 (miR-132-3p) was not significantly regulated. * *p* ≤ 0.05, ** *p* ≤ 0.01, n.s.: not significant.

miR	Fold Change	*p*	
-1-3p	−8.64	0.000	**
-133a-3p	−7.56	0.000	**
-133a-5p	−3.64	0.034	*
-133b	−7.28	0.002	**
-499a-5p	−5.24	0.000	**
-27a-3p	−2.12	0.000	**
-378a-5p	−3.33	0.000	**
-199a-3p	−2.41	0.015	*
-199a-5p	−1.77	0.000	**
-497-5p	−1.78	0.000	**
-15a-5p	−1.48	0.09	n.s.
-18a-5p	−1.59	0.222	n.s.
-19b-3p	−1.7	0.015	*
-132-3p	+1.3	0.275	n.s.
-155-5p	+2.92	0.001	**

**Table 5 biomolecules-13-00884-t005:** Correlations between acute responses of different miRs. The table summarizes strong correlations (|r| > 0.5) between pairs of individual miRs. All correlations were highly significant (*p* ≤ 0.01).

miR	1-3p	133a-3p	133a-5p	133b	499a-5p	378a-5p	497-5p	199a-5p	27a-3p	15a-5p	18a-5p	19b-3p
**1-3p**	-	0.921	0.965	0.942	0.819	0.838						
**133a-3p**	0.921	-	0.947	0.956	0.924	0.878						
**133a-5p**	0.965	0.947	-	0.958	0.981	0.946						
**133b**	0.942	0.956	0.958	-	0.847	0.840						
**499a-5p**	0.819	0.924	0.981	0.847	-							
**378a-5p**	0.838	0.878	0.946	0.840		-						
**497-5p**							-	0.683	0.511			
**199a-5p**							0.683	-				
**27a-3p**							0.511		-			
**15a-5p**										-	0.683	0.511
**18a-5p**										0.683	-	0.767
**19b-3p**										0.511	0.767	-

## Data Availability

All data are available from the authors upon reasonable request.

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
