# Peer review of "Skeletal Muscle MicroRNA Patterns in Response to a Single Bout of Exercise in Females: Biomarkers for Subsequent Training Adaptation?"

_biomolecules, 2023, doi:10.3390/biom13060884_

Round 1
Reviewer 1 Report
The current manusrcipt investigates the acute microRNA response to a single bout of prolonged endurance exercise. The manuscript indicates that the current study population was from a previously described in two previous studies Thiel et al. 2019 and Widmann et al 2022, however, after looking at current and two previous papers, the n described in each is different a more detailed description of the current cohort is needed. Furthermore, there is a lack of a description of the fitness status of the participants as this could have impacted the observed responses. A more detailed description of the cohort and their baseline fitness is necessary and could help explain some of the variance in the data, or identify other potential factors that contributed to the variance. A further explanation as to why the 6 specific samples were included in the current study and if other samples were also analyzed. The analysis seems appropriate, as does the presentation of the results and the discussion.
Author Response
Reviewer #1:
First, we would like to thank the reviewer for helpful suggestions and comments. Our specific responses are listed below.
The current manuscript investigates the acute microRNA response to a single bout of prolonged endurance exercise. The manuscript indicates that the current study population was from a previously described in two previous studies Thiel et al. 2019 and Widmann et al 2022, however, after looking at current and two previous papers, the n described in each is different a more detailed description of the current cohort is needed.
We agree with the reviewer in stating that this might be confusing: Our original goal (as described in Thiel et al., 2019) had been to include n=60 (approximately 50% male and female) in the study. Due to the fact that initially, more female subjects could be recruited, there was a total of n=30 female and n=12 male participants when the COVID19 pandemic led to lockdown of our outpatient clinic and training center. Thus, several studies (specifically Widmann et al., 2022 and the current one) were carried out on the female cohort only, since n=12 for the male cohort appeared too low to reach any statistically sound conclusions with regard to most scientific questions. This is now specified in the text in more detail (p.3, l.37-38 and p.4, l.21-23; redlined version).
Furthermore, there is a lack of a description of the fitness status of the participants as this could have impacted the observed responses. A more detailed description of the cohort and their baseline fitness is necessary and could help explain some of the variance in the data, or identify other potential factors that contributed to the variance.
We agree with the reviewer in stating that a more detailed description of the study cohort, specifically their baseline fitness, would be helpful in the context of interpretation of our results. Thus, we have now included this information in the “Methods” section (p.3, l.38-40; redlined version).
A further explanation as to why the 6 specific samples were included in the current study and if other samples were also analyzed. The analysis seems appropriate, as does the presentation of the results and the discussion.
We agree with the reviewer in stating that this point was not adequately described in the previous version of the manuscript: The 6 subjects included in the screening procedure were the same than those that had been analyzed by microarray analysis for the Widmann et al., 2022, study. They had been chosen based on maximum similarity with regard to gender, age, BMI and baseline fitness to reduce variance of results of the screening approach to a minimum – which, owing to the low n number, we regarded as absolutely necessary. The difference between the Widmann et al., 2022, study and the current paper is that Widmann et al. analyzed correlation of baseline miR patterns and changes during the first training period with changes in V̇O2max, whereas now, we analyzed the miR acute response in connection with a single bout of exercise and training-induced changes in V̇O2max. This is now specified in the text in more detail (p.3, l.50 – p.4, l.2 and p.4, l.11-12; redlined version).
Reviewer 2 Report
The authors aimed to study the effect of a 60 minutes session on a bike ergometer on the expression profile of a number of miRNAs in skeletal muscle.
The paper is well written (although there some flaws) but there are some points that need to be clarified:
The authors indicate that they use muscle samples from 6 healthy controls. It is not until the results section that they mention the training program for these patients (line 174). This should be mention in the methods.
The choice of 60 minutes’ exercise and analysis after 2 hours should be discussed.
The validation in a higher cohort is puzzling. Do the authors mean to say that the validation was performed in muscle samples from the 42 subjects of a previous study mentioned in the first section of the methods? If so, the training regime was different and not comparable with the 6 subjects used for the exploratory part of the study. Also, in the methods it is not mentioned that the 42 subjects were also biopsied.
The study is too descriptive. At least some of the findings should be validated using other techniques. Studying mRNA or protein (immunohistochemistry or Western-Blot) levels of some of the genes targeted by the miRNAs studied would add robustness to the findings.
In the conclusion, the authors state that these kind of study might be a valuable instrument for establishing skeletal muscle miRNA patterns and their modulation by exercise. I believe that the utility of this kind of study should also be further discussed.
Review some grammar flaws
Author Response
First, we would like to thank the reviewer for helpful suggestions and comments. Our specific responses are listed below.
Reviewer #2:
The authors aimed to study the effect of a 60 minutes session on a bike ergometer on the expression profile of a number of miRNAs in skeletal muscle.
The paper is well written (although there some flaws) but there are some points that need to be clarified:
The authors indicate that they use muscle samples from 6 healthy controls. It is not until the results section that they mention the training program for these patients (line 174). This should be mention in the methods.
We agree with the reviewer in stating that the exercise protocols should be mentioned in the “Methods” section and now do so in the revised version (p.3, l.40-50; redlined version).
The choice of 60 minutes’ exercise and analysis after 2 hours should be discussed.
We agree with the reviewer in stating that timing is an important point.
The 60 min duration for the reference training session was mainly chosen for two reasons: (1) It is a typical duration of a training session in recreational sports and in a majority of training sessions in the literature, thus facilitating comparability. (2) It corresponds to the duration (and, at least, approximately, intensity) of an individual training session in our MICT training program. In the iReAct study, a broad variety of parameters, not only of the biochemical / physiological type, was assessed, among others also subjects’ psychological characteristics, e.g. the affective response to exercise. These parameters were not only assessed at reference trainings, but also several times during regular training sessions. A more or less similar design of reference trainings and “regular” training sessions (at least those of the MICT type) also enhanced comparability of the results.
As stated by the reviewer, skeletal muscle biopsies were taken two hours after the end of the reference training. This time point was chosen based on typical kinetics of regulation of mRNAs, miRs, and proteins, and was, of course, a compromise, since due to ethical and also medical considerations, repeated muscle biopsies were not an option. By contrast, blood was drawn twice after the reference training (immediately afterwards and 3h after the end of the reference training).
But of course, as the reviewer states correctly, both timing issues might have affected our results. The “Discussion” section now refers to this point in more detail (p.18, l.34-39; redlined version).
The validation in a higher cohort is puzzling. Do the authors mean to say that the validation was performed in muscle samples from the 42 subjects of a previous study mentioned in the first section of the methods? If so, the training regime was different and not comparable with the 6 subjects used for the exploratory part of the study. Also, in the methods it is not mentioned that the 42 subjects were also biopsied.
We agree with the reviewer in stating that this is confusing and would like to clarify this point: The six (all-female) subjects were the same than the ones chosen for screening in the Widmann et al., 2022, study. qPCR validation was then done with these n=6 plus biopsy material from the remaining n=24 females (or, more exactly, n=23, since for technical reasons, miR samples of one subject could not be included in the analysis). Thus, qPCR validation was carried out with n=29 (6+23) samples of all-female subjects. This is now specified in the text in more detail (p.1, l.24, p.4, l.21-23, p.16, l.5-6, p.17, l.2, p.18, l.2 and p.18, l.27; redlined version).
The study is too descriptive. At least some of the findings should be validated using other techniques. Studying mRNA or protein (immunohistochemistry or Western-Blot) levels of some of the genes targeted by the miRNAs studied would add robustness to the findings.
We agree with the reviewer in stating that validation of miR targets would corroborate our findings. Unfortunately, our fine needle biopsies did not yield enough tissue to perform Western blot or immunohistochemistry analysis in parallel to miR isolation. However, we compared our data with those from a previous study (Schild et al., J. Proteom. 122, 119-132, 2015), also involving our department. In this study, both endurance-trained and untrained participants underwent an acute, 60-min bout of submaximal exercise on a bicycle ergometer, comparable to participants of the current study. However, in this study, proteomics was done on muscle biopsies. With regard to fatty acid metabolism (the main metabolic pathway discovered as target of our set of differentially expressed miRs; both in the current study and in Widmann et al., 2022), the authors found regulation of a broad variety of enzymes, both in trained and in untrained individuals. Interestingly, several of the respective genes are targets of the differentially expressed miRs identified in our study, e.g. the HADH (hydroxyacyl-coenzyme A dehydrogenase, mitochondrial) gene, encoding a central player of b-oxidation, which is a major target of myomiR-1-3p. And, of course, in the future, more refined studies encompassing different methods and assessing RNA, miR and protein regulation in the same set of samples, will have to be carried out. This is now also mentioned in the text and the Schild et al., 2015, paper is cited as a reference (p.19, l.45 – p.20, l.3; redlined version).
In the conclusion, the authors state that these kind of study might be a valuable instrument for establishing skeletal muscle miRNA patterns and their modulation by exercise. I believe that the utility of this kind of study should also be further discussed.
As suggested by the reviewer, we now refer to utility of our data from a practical perspective in more detail (p.21, l.1-7; redlined version).
Comments on the Quality of English Language
Review some grammar flaws
As suggested by the reviewer, we have thoroughly revised the complete manuscript and corrected some inconsistencies and flaws. In one case, I am not completely sure whether the wording is correct (labelled in yellow).
Round 2
Reviewer 1 Report
N/A
Reviewer 2 Report
The authors have addressed my concerns